# Flexural Capacity of Fire-Affected Concrete Members with Recycled Glass Aggregate and Glass Pozzolan

**Enrique Gonzalez Tapia [1,\*] and Nur Yazdani [2]**

1   M.S., E.I.T, Civil Engineer, Arlington, TX 76019, USA
2   Department of Civil Engineering, University of Texas at Arlington, Arlington, TX 76019, USA
\*   Correspondence: enrique.gonzalez@mavs.uta.edu

**Abstract:** Adding recycled glass components to concrete mixes is a novel and sustainable option. Prior studies have recommended replacement ratios of 30% glass aggregates and 20% ground glass pozzolan as optimum substitutions in concrete mixes. The compressive strength of glass concrete has been shown to increase when glass components are used with these proportions. Less information is available on the flexural strength of such concrete, and no previous research has been conducted on the flexural capacity of glass concrete exposed to high temperatures. To bridge this knowledge gap, cured concrete cylinders and beam samples using various glass coarse aggregate and ground glass pozzolan replacement ratios were heat-treated in a furnace. The heat-affected samples were then tested for their compressive and flexural capacities. It was found that glass pozzolan increased the mix workability, while glass aggregates reduced it. The compressive strengths were modesty increased and the flexural capacities were drastically reduced (up to 93%) after heat exposure. Therefore, recycled sustainable glass concrete may be efficiently used in concrete compressive members and in properly designed flexural members. It can also be used efficiently in architectural non-load-bearing members for insulating and aesthetic effects.

**Keywords:** recycled glass; flexural capacity; thermal effect; compressive strength; pozzolan; coarse aggregate

## 1. Introduction

The concrete construction industry is continually evolving by adopting new technology, methods, and materials. In a world with an increasing population and demands for improved infrastructure, the need for better and more sustainable materials grows as well. This need has inspired researchers to find ways to recycle and reuse various products. Concrete is widely used in construction for almost all infrastructure types, such as roads, bridges, and buildings. It is very flexible in application and provides a high structural capacity. However, cement production for concrete is a major source of environmental pollution and accounts for 5–10% of the $CO_2$ production worldwide [1]. Fortunately, several materials, such as fly ash and silica fume, byproducts that are recycled from other production processes, can be added to the concrete mixture as partial cement replacements to reduce the pollution from the cement production process.

Some recent research has focused on adding recycled glass to concrete mixtures. Glass is commonly used in containers, windows, automobile parts, and other applications. However, after the useful lives of the glass products are over, most of the glass ends up in landfills. In the U.S., about 9.4 million tons of glass are generated annually, with about two-thirds of it ending up in landfills and only one-third being recycled [2]. Glass can be recycled and reused by retrieving it from landfills or recollection centers and sending it to processing plants where it is crushed into smaller pieces. It is then melted and new products may be made in a process that is sustainable. A possible problem with using glass in concrete is the alkali–silica reaction (ASR), forming a gel that can create voids inside

concrete. The ASR process can be mitigated by using admixtures. The use of recycled glass benefits the environment by reducing landfill volumes and improving the structural properties of concrete by increasing its compressive strength [3]. Additionally, since the unit weight of glass is smaller compared to that of concrete, the self-weights of members can be reduced, thus increasing the design/construction economy.

Recycled glass aggregates (RGAs) can be added to concrete as aggregates or as a pozzolan. When used as an aggregate, glass can be a replacement for both fine and coarse aggregates. Different replacement ratios have been used in the past to find the optimum glass aggregate replacement ratios, and several studies have been conducted to analyze the effect of replacement ratios on the concrete's compressive strength. Using coarse glass aggregate as a one-third replacement of the natural coarse aggregate in concrete has been found to be optimal [4]. This coarse aggregate replacement ratio can be found in several studies. However, replacement proportions exceeding 30% negatively affect the concrete compressive strength [3]. Research on the effect of RGAs on the flexural strength of concrete has also been limited. It was reported that the addition of RGAs generally reduces the flexural strength of concrete as the RGA ratio increases [5,6]. It was also found that RGAs as partial coarse aggregate replacements were more beneficial in enhancing the concrete flexural strength than fine aggregate replacements [6]. As the literature suggests in general, a decrease in the flexural strength can be expected depending on the type of glass aggregates used in the concrete mixtures.

Recycled glass (RG) used as a partial pozzolanic replacement in concrete mixes is a promising development. When glass is ground to a particle size below 300 μm, it develops pozzolanic properties and is a sustainable replacement of the cement in concrete. The reaction occurs when the glass particles encounter the lime and water present in concrete mixes. Below a particle size of 100 μm, ground glass (GG) can have a pozzolanic reactivity greater than that of fly ash [1]. At 28 days, samples with 10%, 20% and 30% cement replacement with GG showed a lower compressive strength than that of control samples [7]. However, GG enhanced the compressive strength as it aged beyond 28 days. The compressive strength of concrete with a 20% cement replacement with GG was higher than the control samples after 56 days [2]. A cement replacement of about 20% with GG is recommended as an optimal ratio [2,8]. As with compressive strength, research on flexural performance has shown a slight reduction with the addition of GG [9]. However, at a 20% cement replacement ratio, the concrete flexural strength increased with age after 28 days [2].

The effect of elevated temperature on the mechanical properties of concrete with recycled glass has been studied previously. A wide range of parameters were used, including different proportions of glass, types of recycled glass and temperature ranges. No previous research has been conducted on flexural strength after heat exposure. Concrete with a 21% addition of GG better retained its compressive strength after an exposure to fire [10,11]. The compressive strength of concrete with glass aggregates decreased as the exposure temperature increased. On the other hand, the samples containing 10% fine glass aggregate, coarse glass aggregate and a combination of both showed a higher compressive strength compared to the control samples up to 700 °C [12].

When used as an aggregate replacement, the alkali–silica reaction (ASR) can be a potential problem in concrete with partial RGA replacement. This reaction is surface-dependent; therefore, bigger-size coarse and fine aggregate particles can lead to the ASR [1]. The problem can also occur in non-glass concrete if the aggregate contains certain siliceous rocks and minerals, such as opaline chert, stained quartz and acidic volcanic glass [13]. However, the ASR is more prevalent in glass concrete and mitigation actions are recommended [14]. This includes using additives, such as fly ash [7,15]. Careful selection of the type of glass used in concrete can decrease the possible ASR effect. Using 300 μm or less GG as concrete pozzolan can minimize the ASR [1].

Glass generally softens and becomes malleable enough to work with when it reaches a temperature of about 700 °C [16], so the effect of such temperatures on glass concrete and its relevant properties are of high interest. The Background Review section presented

in this paper shows that very limited research has been undertaken to date on the overall mechanical properties of glass concrete, and especially on the flexural capacity of glass concrete members. Additionally, the separate effects of adding recycled GG as pozzolan and recycled CA and FA to concrete mixes for flexural members under heat exposure are also not well understood. This paper bridges this important knowledge gap through experimental testing and a data analysis of concrete beam samples with recycled glass components. This will pave the way for further evaluations and eventually might lead to the acceptance of glass concrete by the concrete industry with confidence. This "green" concrete will be beneficial to the environment by reusing a waste product, reducing the demands on landfills and reducing the $CO_2$ emissions from cement production.

## 2. Sample Preparation

Three concrete mixes were devised to investigate the flexural capacity of concrete beams with different kinds of recycled glass additives. The replacement ratios were based on results from previous studies [2,4,8]. The three concrete mixes are as follows with associated nomenclatures:

- Plain concrete: PC;
- Concrete with GG as 20% cement replacement only: GP;
- Concrete with GG as 20% cement replacement and CGA as 30% coarse aggregate replacement: GP + CGA.

A total of 18 beams were prepared, with six beams for each mix (200 × 200 × 914 mm), as shown in Table 1. For the experimental design, a total of nine parametric combinations with two sample replications each was used. This resulted in twelve PC beams, six GP beams and six GP + CGA beams. Three 100 × 200 mm cylinders were also cast separately for each mix for compressive strength determination. A diagram presenting the beam dimension is shown in Figure 1. No reinforcement was provided in the beams in order to evaluate the load capacity of the concrete only. To track the actual temperature during heat treatment, a mineral-insulated, ungrounded thermocouple was placed at the midspan of each beam sample with 100 mm embedment during concrete casting. Type K thermocouples from DURO-SENSE Corporation, model number MTC-D-12521-G-42-SP-A, were used.

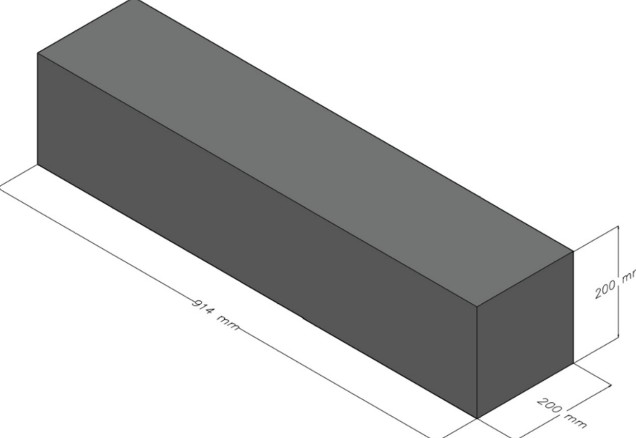

**Figure 1.** Beam Dimensions.

For typical fires, the fire department's response time depends on several factors, such as the relative location of the incident. Depending on the situation, the time to extinguish a fire can range from 30 min to 1 h [17]. Following this timeframe, two heat exposure times of 30 min and one hour were used in the test formulation.

The ACI volumetric method (1985) was used to select the concrete mix design. The target 28-day concrete compressive strength was 24.13 MPa. The water-to-cement ($w/c$) ratio was set as 0.51 for all samples. The mixing proportions were calculated based on these

initial determinations, and the calculations were performed for one cubic meter of concrete. Since the volumetric method was used, the amount of cement replaced was calculated by volume instead of by weight. Since glass has a lower unit weight than concrete, the weight of the volume of glass needed to replace the same volume of cement was slightly lower. The same procedure was used to determine the CGA replacement. The mix design for the three types of beams is presented in Table 2.

**Table 1.** Experimental Matrix.

| No. | 20% GG Replacement | 30% CGA Replacement | Heat Treatment (Hours) | No. of Beams | Mix Type |
|---|---|---|---|---|---|
| 1 | Yes | No | No | 2 | GP |
| 2 | No | No | No | 2 | PC (control) |
| 3 | Yes | No | 0.5 | 2 | GP |
| 4 | No | No | 0.5 | 2 | PC |
| 5 | Yes | No | 1 | 2 | GP |
| 6 | No | No | 1 | 2 | PC |
| 7 | Yes | Yes | No | 2 | GP + CGA |
| 8 | Yes | Yes | 0.5 | 2 | GP + CGA |
| 9 | Yes | Yes | 1 | 2 | GP + CGA |
| | | | Total | 18 | |

**Table 2.** Mix Design for 1.0 m$^3$ of PC and Glass-Added Beams.

| Ingredient | Weight (kg) | | |
|---|---|---|---|
| | PC Beams | GP Beams | GP + CGA Beams |
| Water | 185 | 185 | 185 |
| Cement | 344 | 291 | 291 |
| Regular coarse aggregate | 1033 | 1033 | 715 |
| Fine aggregate | 848 | 848 | 848 |
| Ground glass | None | 53 | 53 |
| Coarse glass aggregate | None | None | 291 |

The materials were weighed out in the corresponding proportions and added to the mixer. A mixing time of two minutes was allowed to achieve a good consistency. The concrete was then placed in plywood formwork with assistance from a hand-held vibrator. A slump test and air content test were performed with the fresh concrete following ASTM procedures [18,19]. The samples were continuously cured through the application of a curing compound on all exposed surfaces and covered to avoid direct exposure to sunlight. The completed samples are shown in Figure 2.

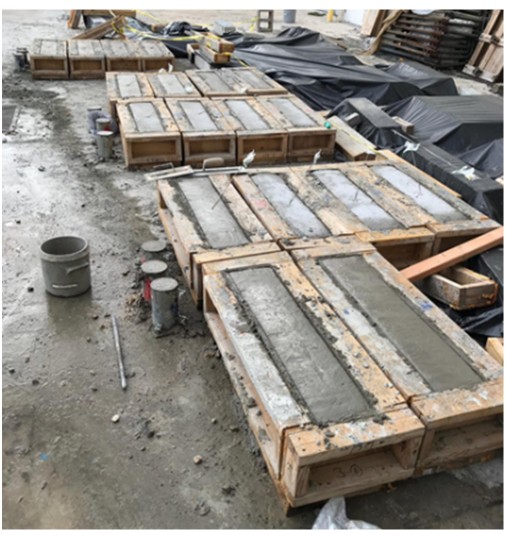

**Figure 2.** Cast Beam Samples and Cylinders.

### 3. Fire Test Setup

After curing, the samples were heat-treated following the ASTM E119 [20] standard. An electrical furnace was used to perform the heat treatment. The furnace was equipped with a controller to configure the temperature. The furnace software was programmed to follow the ASTM E 119 temperature curve, as shown in Figure 3.

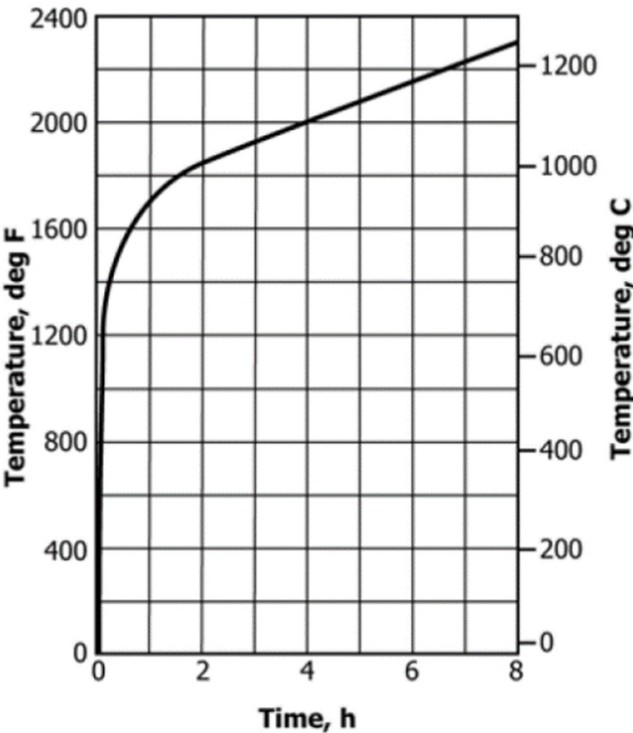

**Figure 3.** Time–Temperature Curve [20].

To record the exact temperatures inside the furnace and to have a better understanding of the heat distribution on the sample surface, two additional thermocouples were used on the top and bottom surfaces of each sample. Temperature data were automatically collected at a rate of 10 readings per second. Figure 4 shows the placement of a sample inside the furnace and the thermocouple setup. Heat-treated samples were allowed to cool to room temperature before their removal from the furnace. The cylinders were heat-treated similarly, except that no thermocouples were utilized.

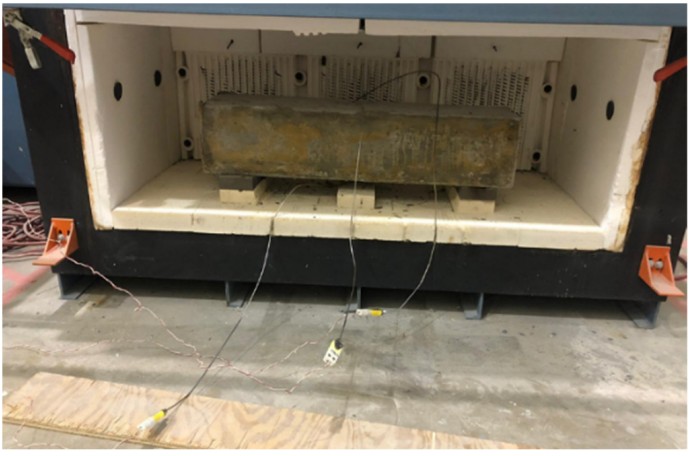

**Figure 4.** Heat Testing Setup.

## 4. Compressive Strength Test

The cylinder compressive strength test was performed in a universal testing machine following the ASTM C39 [21] standard. Due to a scheduling conflict at our structures' laboratory, the samples were tested at 90 days instead of 28 days. The sample ends were capped with sulfur prior to testing [22]. The load was applied at a rate of approximately 1.78 KN per second following the ASTM C39 [21] specifications, as shown in Figure 5. The average value from the three cylinders was reported as the compressive strength of concrete.

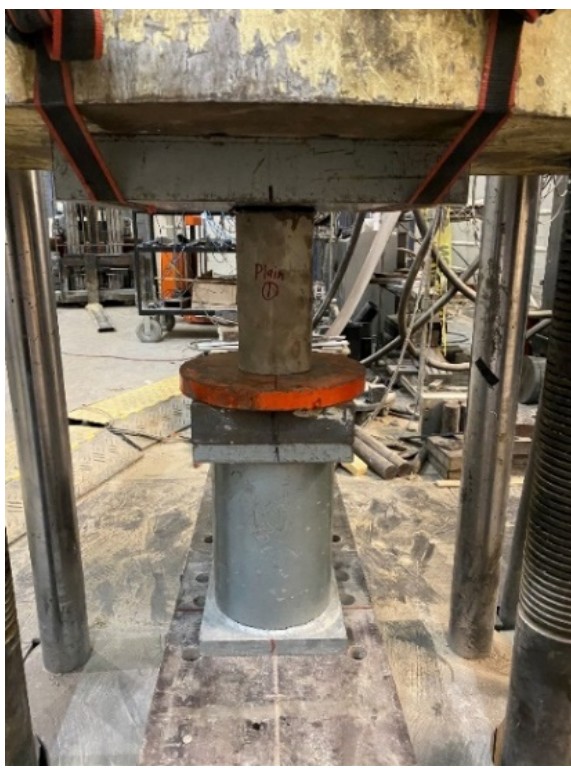

**Figure 5.** Cylinder Compressive Strength Test Setup.

## 5. Residual Flexural Strength Test

A 4-point setup was used for the residual flexural test setup of the heat-affected beam samples [23]. The test was performed in the universal testing machine. Simple supports were simulated through steel rods placed below the concrete beam samples. The 4-point load was applied through a load cell of 2000 KN capacity, a stiff steel beam at top of the concrete beam and two roller supports, as shown in Figure 6. Two linear varying deflection transducers (LVDT) were installed on each side of the concrete beam to measure maximum vertical deflections. A rigid aluminum bar attached to the concrete beam served to connect to the LVDTs. Foil strain gages were attached to the top and bottom of the beam to measure maximum compressive and tensile flexural strains. The LVDT and strain data were automatically gathered through a DAQ box and a laptop.

The load was added using a universal testing machine with a 1778 KN capacity. The samples were continuously loaded at a rate of approximately 0.2 KN/sec during testing. The cracking and failure modes were monitored by visual inspection. Since a brittle failure was expected due to the lack of steel reinforcement, no marks on the beam surface to monitor crack propagation could be made due to safety concerns. After failure, the rupture location and any additional cracks were noted once the beams were removed from the testing machine.

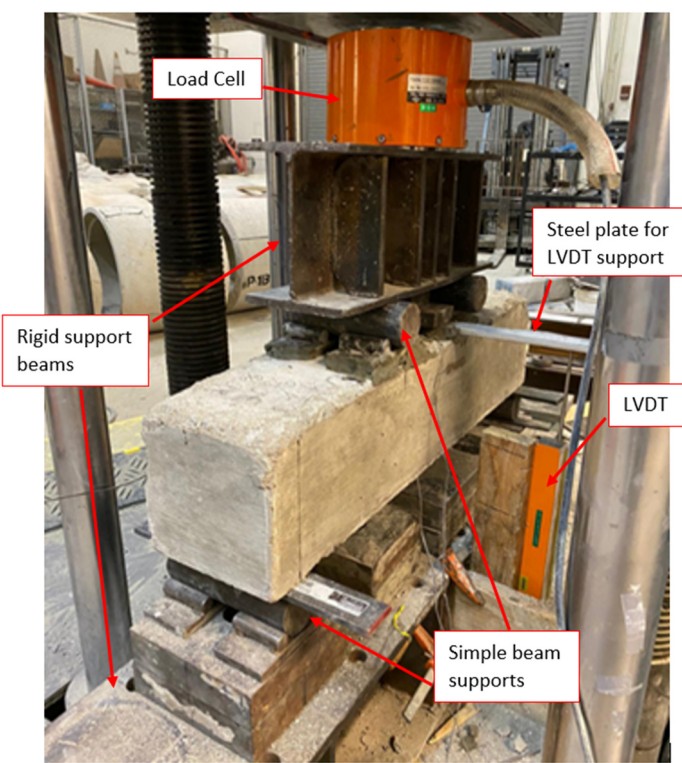

**Figure 6.** Flexural Test Setup.

## 6. Results and Discussion

### 6.1. Fresh Concrete Results

Table 3 presents the average test results from the fresh concrete testing. The highest slump was observed in the GP mix, showing that ground glass pozzolan improves the workability of concrete. The GP + CGA mix decreased the slump below that from the control mix, as also can be seen from the stiffness of the mix during the concrete mixing time. As suggested by the literature and observed during the mixing process, the glass did not absorb as much water as the natural aggregates [1]. However, it was noted that the more jagged the glass aggregate geometry, the lower the concrete workability. Additionally, since the largest available size for the CGA was only 16.13 mm, the finer aggregate tended to decrease the workability of the mix.

**Table 3.** Fresh Concrete Test Results.

| Mix | Slump (mm) | Air Content (%) | Temperature (°C) |
| :---: | :---: | :---: | :---: |
| PC | 165 | 1.8 | 28.3 |
| GP | 175 | 1.4 | 30.0 |
| GP + CGA | 146 | 2 | 32.8 |

The highest air content was found in the GP + CGA mix while the lowest was found in the GP mix. The concrete casting was performed on a warm day, with the ambient temperature ranging between 26 and 34 °C. The concrete temperatures followed the same trend as the ambient temperature. The plain concrete mix was prepared and cast first during the cool morning environment, followed by the GP and the GP + CGA mixes, respectively, in the afternoon. Table 3 shows that the measured fresh concrete temperatures consistently followed the ambient values.

### 6.2. Compressive Strength Results

The average 90-day concrete compressive strength results from the cylinder testing are presented in Figure 7. It is apparent that the addition of glass increased the compressive

strength of the concrete, although at modest levels. The GP + CGA samples showed the highest strength value at 29.5 MPa. The GP and GP + CGA mixes resulted in 6% and 7.45% strength increases over the control samples, respectively. Although some concerns may exist about the effects of the smooth surfaces of glass aggregates on concrete bonding and strength, the improved compressive strength resulted from the sharp geometric shapes. These shapes allowed for the better stacking of the material inside the concrete, which resulted in more compact samples.

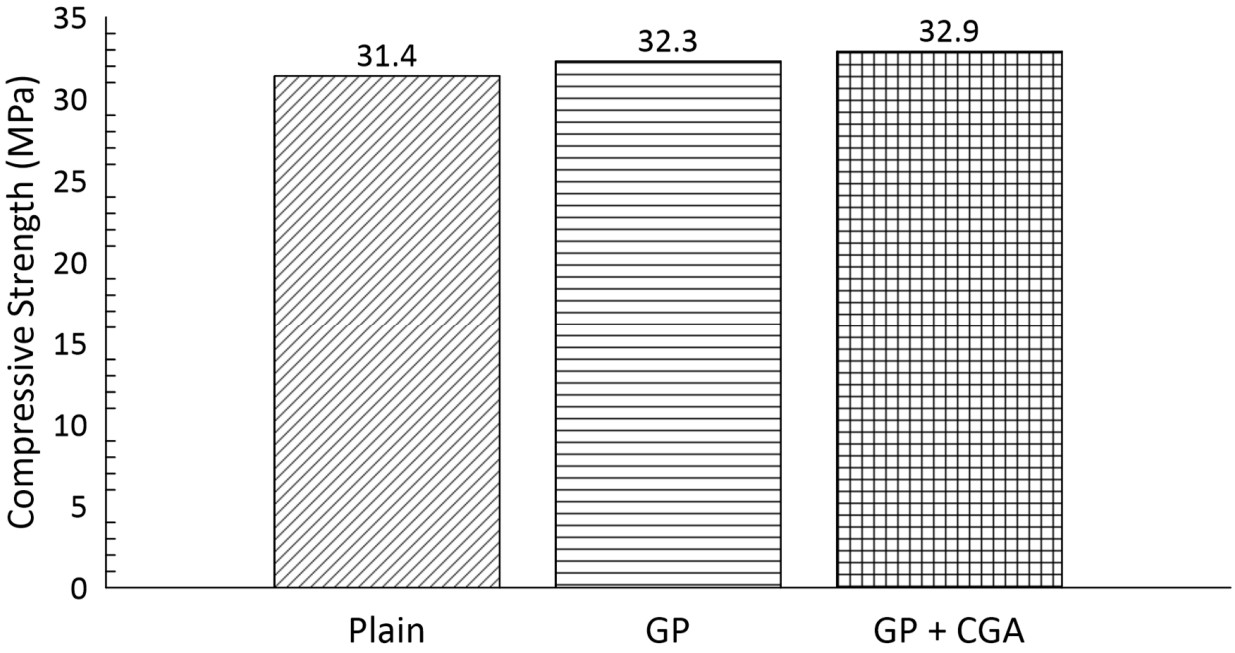

**Figure 7.** Average Compressive Strengths.

The ASTM C39 [21] standard's defined cylinder failure types were used to label the different failure patterns observed in the tested cylinders, as shown in Figure 8. Type 2 failure was observed in the control and GP samples with well-defined cones visible at the bases. The failure patterns in the GP + CGA samples did not resemble any of the ASTM C39 failure types and were characterized by a single diagonal crack from the top surface to the side of the cylinder. These were similar to the Type 4 ASTM fractures; however, additional cracks were observed near the bottom surface with no cone formation.

*6.3. Fire Test Results*

Although the temperature inside the furnace was set to reach 926 °C, the temperatures inside the concrete beams, as measured by the thermocouples, were much less and certainly below the melting point of glass, as shown in Figure 9.

The PC beams with no glass reached higher temperatures during the first 45 min of heat treatment. After the first 30 min of heat exposure, the internal temperature in the plain concrete samples was 10 °C higher than that of the glass samples. This trend continued until 45 min of heat treatment, showing the insulating effect of the addition of glass. After about 45 min, however, the beams containing only GP showed an increase in their internal temperature. The GP + CGA beams experienced their lowest internal temperature after 45 min of exposure. The air content in these samples was also higher, which may also have added to the heat insulating effect. The additional air content and the addition of glass did not affect the compressive strength of the GP + CGA concrete samples. The glass seems to function as an insulator while maintaining an acceptable and improved compressive strength capacity.

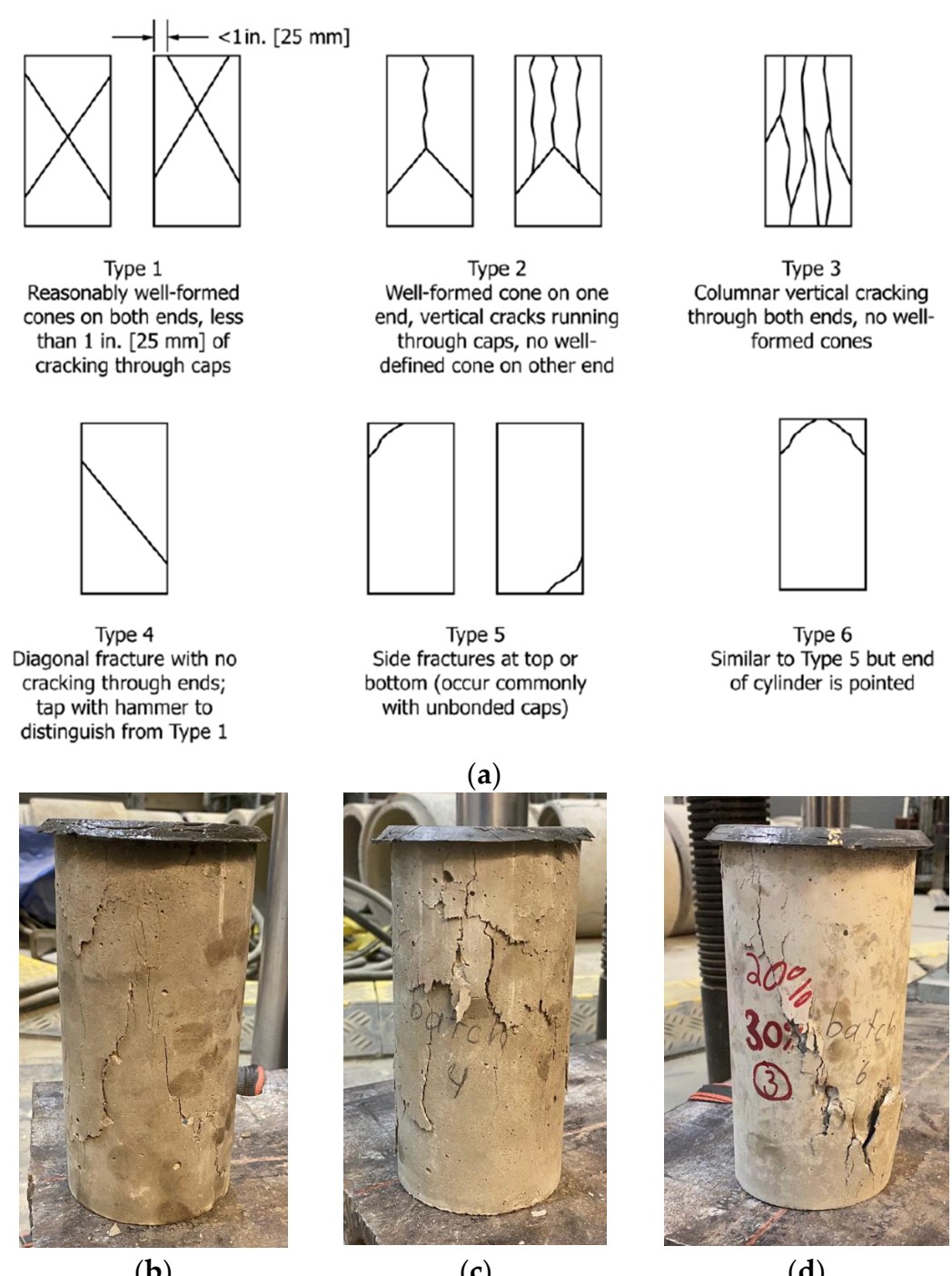

**Figure 8.** Cylinder Failure Types: (**a**) ASTM C39 [21] Fracture Patterns; (**b**) Plain Concrete, Type 2; (**c**) GP, Type 2; and (**d**) GP + CGA, No Specific Type.

### 6.4. Residual Strength Test Results

The residual flexural capacities are expressed in this study as the maximum concentrated mid-span load capacity of each concrete sample during the bending test. For samples with no heat treatment, the GP + CGA beams had the lowest strength, with an average of 47.5 KN (Figure 10). This was due to the smooth surface of the CGA culets, which affected the bonding of the material to the concrete. It may be noted that the GP + CGA samples had the highest compressive strength. This contradiction is evidence that the smooth surface of the CGA did not affect the compressive behavior. When tensile stress was present, however, although a better internal stacking of the material could be achieved, the smooth surface

of the glass affected the bonding of the materials and decreased the tensile capacity of the beams at a regular temperature. However, when only GP was added, the flexural capacity increased to 65.9 KN, which is a 10% increase over the control plain concrete beams. The failure was rapid and brittle for all samples with the fracture at the midspan, as shown in Figure 11.

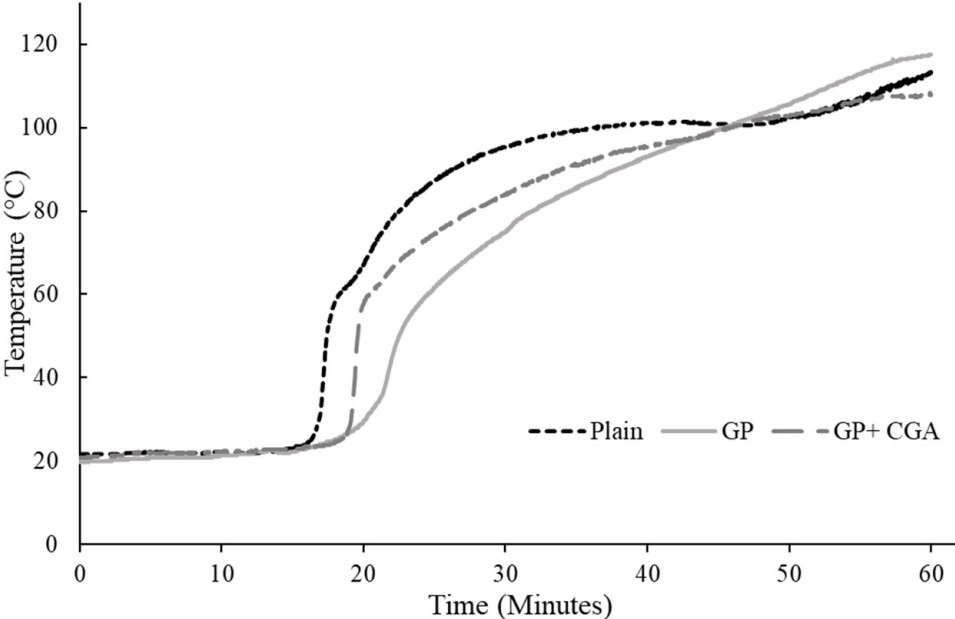

**Figure 9.** Average Internal Temperatures.

The heat treatment resulted in drastic reductions in the flexural capacities with similar trends as those observed for the no-heat samples, as seen in Figure 12. The flexural capacity in the GP + CGA samples was 93% less than that in the GP samples. Interestingly, the failure mode was ductile in the glass samples with the appearance of flexural cracks (Figure 13). This was an unexpected outcome of the test, since no reinforcement was provided. Because the CGA softened a bit within the beams, they added to the ductile behavior after the heat exposure. The GP samples gained ductility while maintaining the highest load capacity.

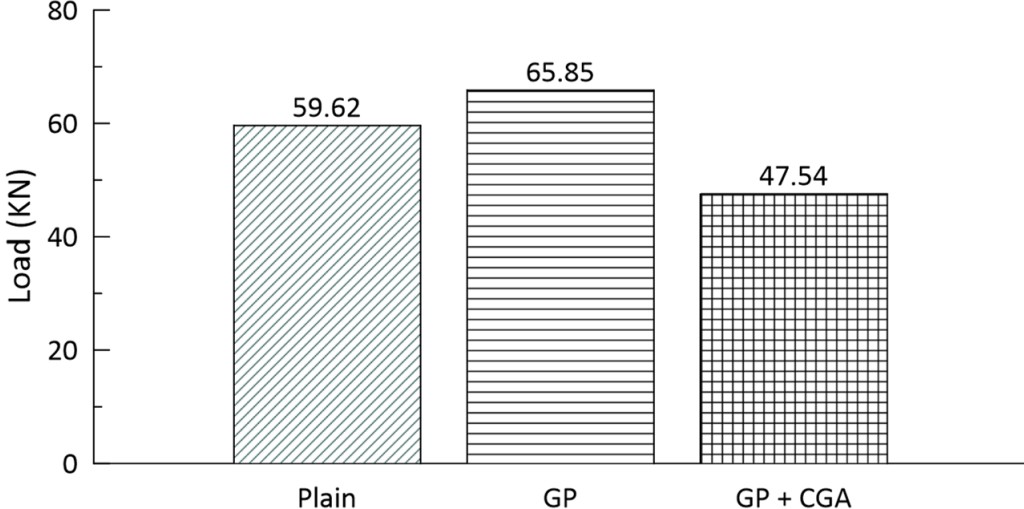

**Figure 10.** Average Flexural Capacity Results; No Heat Exposure.

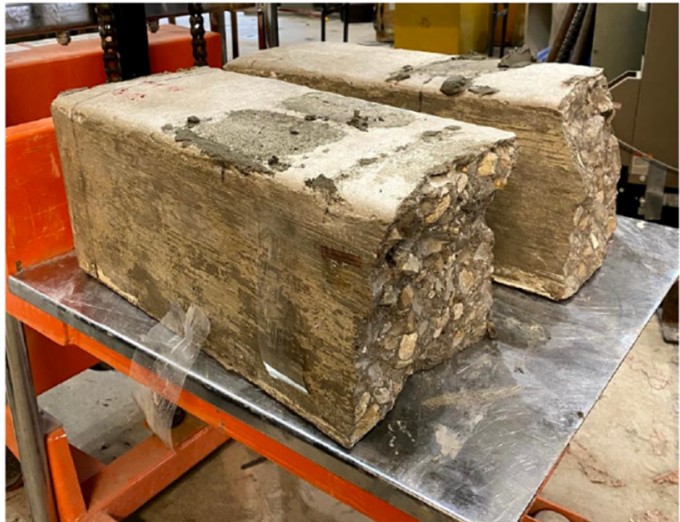

**Figure 11.** Failed Beam Fracture.

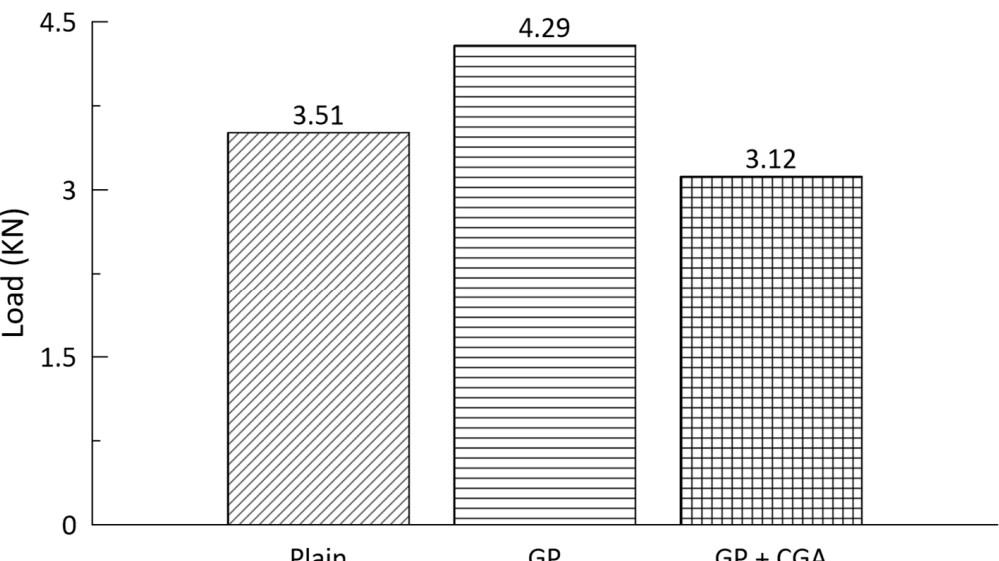

**Figure 12.** Average Flexural Capacity Results; 30-min Heat Exposure.

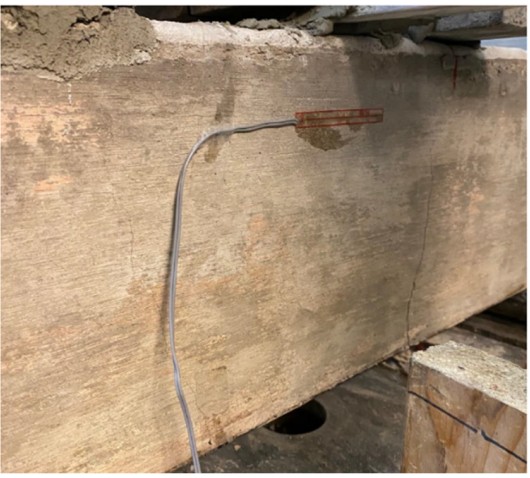

**Figure 13.** Beam with 30-min Heat Exposure Showing Flexural Crack at Failure.

The one-hour heat treatment caused a further reduction in the flexural capacities. The control plain concrete beams became very brittle after the heat treatment and developed large cracks under their own weights, as shown in Figure 14. Therefore, these beams were not tested for bending.

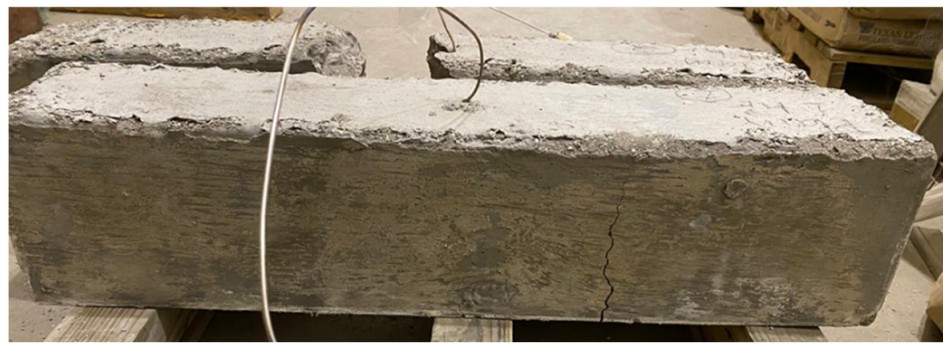

**Figure 14.** Failed Beams before Testing.

As in the previous cases, the GP samples showed the highest load capacity, followed by the GP + CGA samples (Figure 15). However, the relatively low flexural capacities for these samples were evident because they failed in a brittle mode as soon as the testing machine applied the flexural load on the samples.

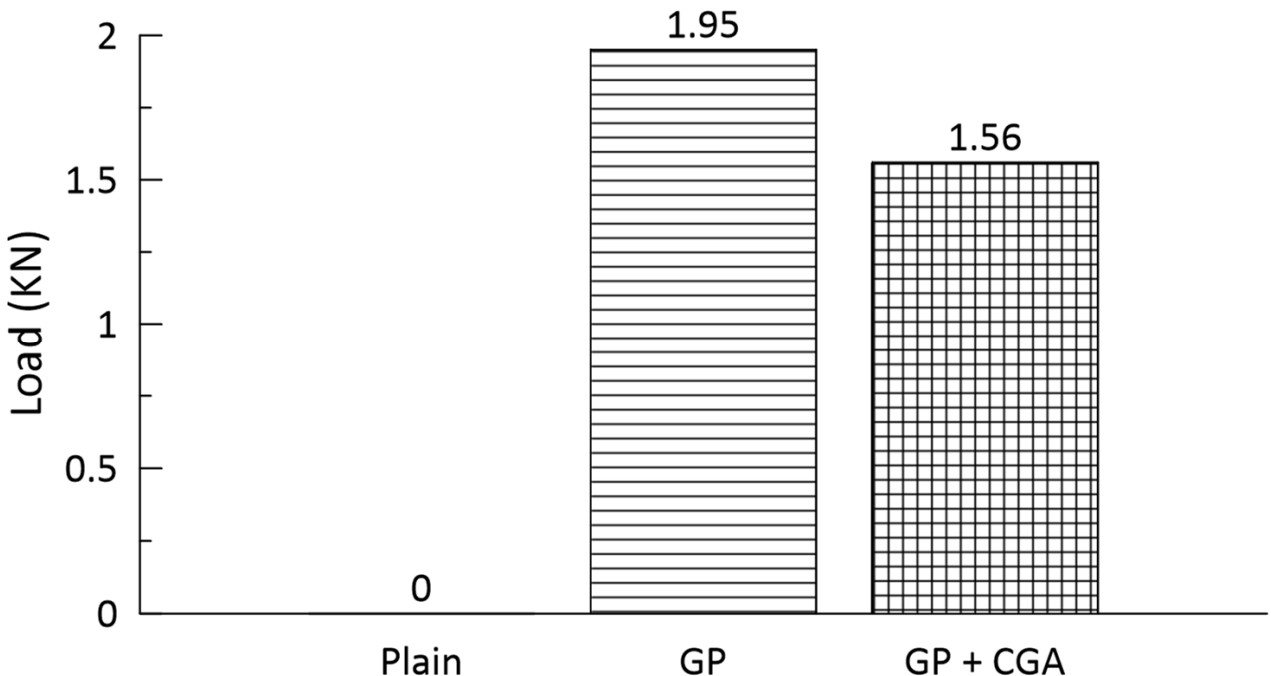

**Figure 15.** Average Flexural Capacity Results; One-Hour Heat Exposure.

The load vs. strain and load vs. displacement graphs were developed for the strongest beams in each category to better understand their flexural behavior. The raw data were modified using the moving average of the data points. The graphs for the beams with no fire exposure are presented in Figures 16 and 17, respectively. All the samples reached a strain of approximately 140 $\mu\varepsilon$ at their time of failure. However, the GP + CGA beam had the lowest load capacity as compared to that of the other beams and also underwent the largest displacement, exceeding 1.016 mm. Although the GP beam had the highest load capacity, its ductility was the lowest.

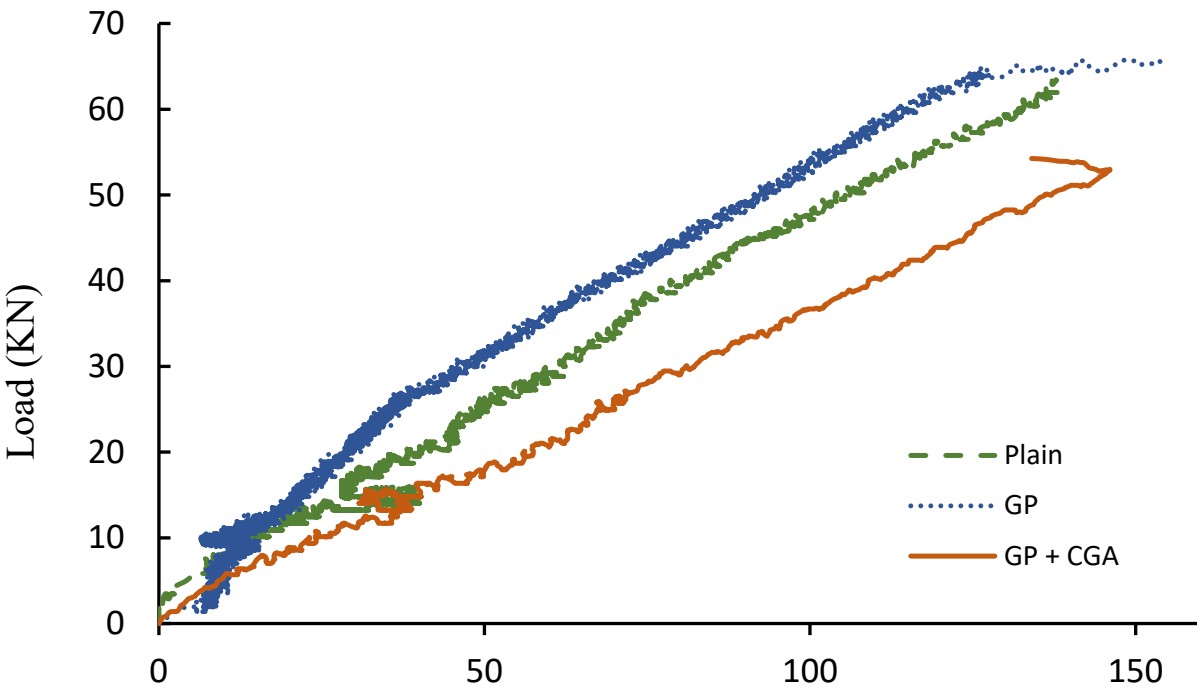

**Figure 16.** Load vs. Strain Variations; No Heat Exposure.

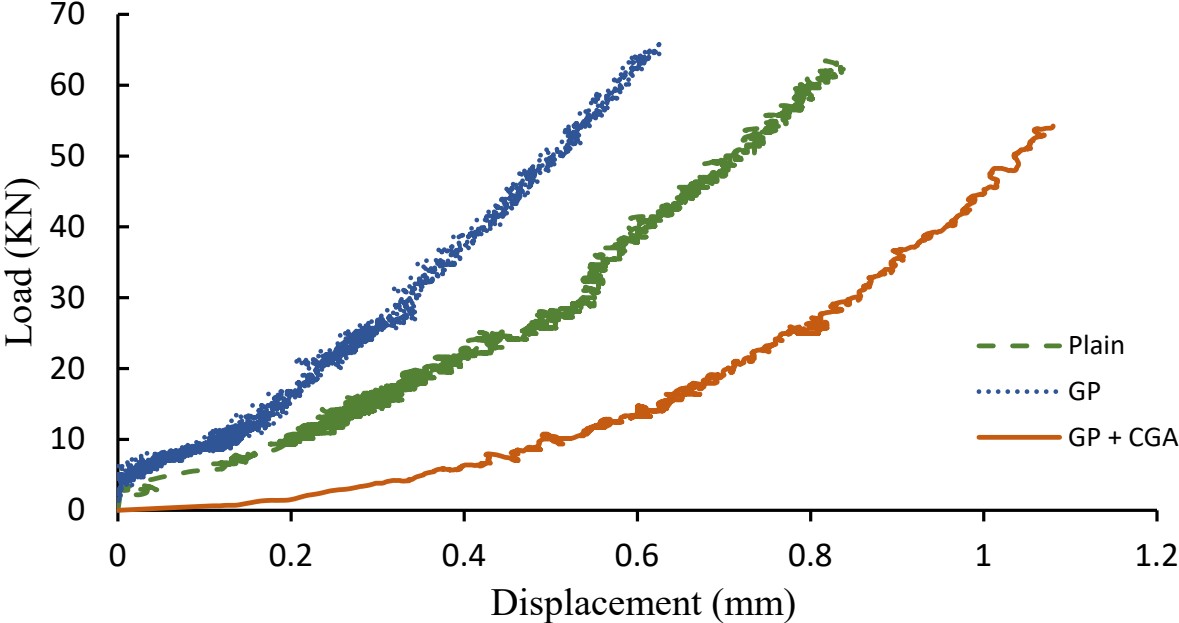

**Figure 17.** Load vs. Displacement Variations; No Heat Exposure.

The load vs. strain and load vs. maximum displacement graphs for the beams with the 30-min heat exposure are presented in Figures 18 and 19, respectively. The GP beam experienced a large strain and the deflection showed a significant increase in the strain compared to that of the beams with no heat exposure. On the other hand, the beam with 20% GP and 30% CGA showed a reduction in the strain compared to the beam with the no-heat treatment. The beam with plain concrete showed similar results to the plain concrete beam with the no-heat treatment. Regarding the displacement, the 20% GP and the plain concrete beams showed similar results of about 0.25 mm. The displacement of the 20% GP and 30% CGA samples was lower than that of the other samples, only 0.07 mm.

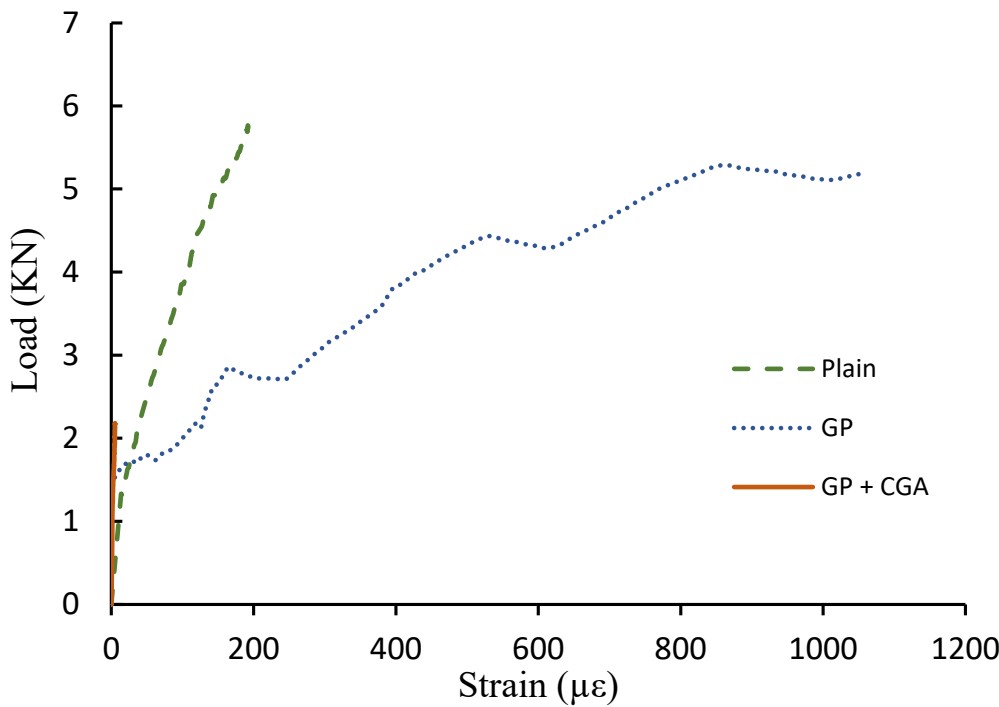

**Figure 18.** Load vs. Strain; Beams with 30-min Heat Treatment.

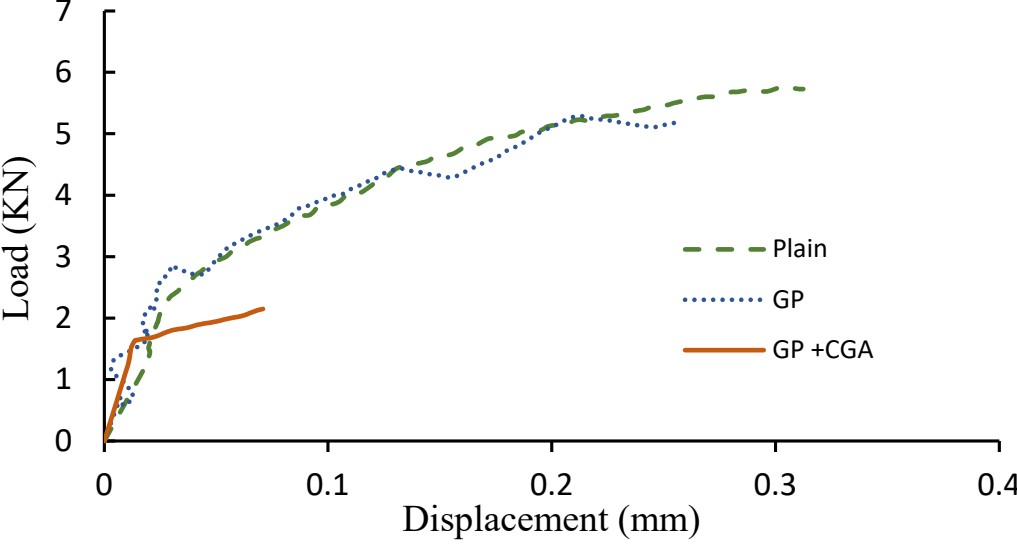

**Figure 19.** Load vs. Displacement; Beams with 30-min Heat Treatment.

The graphs for the beams with one hour of heat exposure were developed. Since the beams failed as soon as the testing machine started applying the load, only a few data points were taken. Due to the small amount of data acquired during this test, the graphs do not show their typical curve pattern. Thus, it was decided that the graphs did not show relevant information, so they are not shown in this paper. However, it was noted that an overall reduction in the strain and the displacement was seen for these kinds of samples when compared to the samples with less heat exposure.

## 7. Conclusions and Recommendations

The following conclusions and recommendations can be made based on the findings from this study:

1. The highest slump was observed in the fresh concrete mix with ground glass pozzolan (GP), showing that it improves the workability. However, the slump in the mix with

GP and the coarse glass aggregates (GP + CGAs) appeared to be quite stiff during mixing and was lower than that in the control mix with no glass added. This was also seen from the stiffness of the mix during mixing. Glass aggregates do not absorb as much water as natural aggregates in concrete. The jagged geometry of CGAs also reduced the concrete workability. Additionally, since the largest available CGA size was only 16 mm, the finer aggregates tended to decrease the workability.

2. The highest air content occurred in the GP + CGA mix while the lowest was in the GP mix. The fresh concrete temperature followed the same trend as the ambient temperature.

3. The addition of glass increased the concrete compressive strength at modest levels. The GP and GP + CGA additions resulted in 6% and 7.45% strength increases over those of the control samples, respectively. Although CGAs have generally smooth surfaces, any negative effects on the bonding and strength are more than made up for by the relatively sharp and jagged aggregate shapes.

4. ASTM-defined Type 2 failures were observed in the control and GP samples with well-defined cones. The failure patterns in the GP + CGA samples did not resemble any of the ASTM failure types and were characterized by a single diagonal crack from the top surface to the side of the cylinder. It was similar to the Type 4 ASTM fracture; however, additional cracks were observed near the bottom surface with no cone formation.

5. The temperatures inside the samples were much less than the surface and well below the glass melting point. This indicated that the glass additives did not melt and were intact within the concrete matrix, functioning as an insulator while maintaining an improved compressive strength.

6. For samples without a heat treatment, the GP + CGA beams had the lowest strength, with an average of 47.5 KN. This was due to the smooth surface of the CGA culets, which affected the bonding. The GP + CGA samples had the highest compressive strength. This contradiction is evidence that the smooth surface of the CGA does not affect the compressive behavior. Under tensile stress, the smooth surface of the glass affects the bonding of the materials and decreases the tensile capacity of the beams at a regular temperature. However, when only GP was added, the flexural capacity increased to 65.9 KN, which is a 10% increase over that of the plain concrete beams. The failure was rapid and brittle for all samples with a fracture at the midspan.

7. The heat treatment resulted in drastic reductions in the flexural capacities with increasing strength losses at higher levels of heat exposure. The flexural capacity in the GP + CGA samples was 93% less than in that in the GP samples with ductile failure modes and flexural cracks. The GP samples gained ductility while maintaining the highest load capacity.

8. The GP + CGA samples showed the largest ductility in terms of strain and displacements, while the GP samples showed the lowest ductility.

9. Glass aggregates affect the flexural strength of concrete. Although they increase the concrete compressive strength, they reduce tensile strength in flexure. Thus, these aggregates may be used in structural flexural applications with adequate design considerations. Due to their insulating effects, they can be effectively used in architectural applications, such as in non-load-bearing panels. Additionally, glass aggregate additives may add an aesthetic appeal in such applications.

**Author Contributions:** Conceptualization, N.Y.; Methodology, N.Y.; Formal Analysis, E.G.T.; Investigation, E.G.T.; Resources, N.Y.; Data Curation, E.G.T.; Writing—Original Draft Preparation, E.G.T.; Writing—Review and Editing, N.Y.; Visualization, N.Y.; Supervision, N.Y.; Project Administration, N.Y.; Funding Acquisition, N.Y. All authors have read and agreed to the published version of the manuscript.

**Funding:** Funding for this study was provided by the Texas Department of Transportation.

**Data Availability Statement:** The data presented in this study are available on request from the corresponding author. The data are not publicly available due to privacy issues.

**Conflicts of Interest:** The authors declare no conflict of interest.

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
