# Peer review of "Flexural Capacity of Fire-Affected Concrete Members with Recycled Glass Aggregate and Glass Pozzolan"

_fire, doi:10.3390/fire5060207_

Round 1
Reviewer 1 Report
The article brings a great contribution to the material science research. Therefore, the following observation are done on the text:
- Figure 3: It is suitable to show, in the same graph, the time-temperature curve of the thermocouple used to monitor the furnace interior temperature.
- Review lines 377 and 378, mentioning if the conclusion (8) refers to ambient temperature or high temperature condition. If it is about high temperature, such conclusion is different from the behaviour commented in lines 321 to 329.
- Review the sentence in lines 322 – 323.
- Write chapter 7 and References in the same format of the template.
Author Response
The article brings a great contribution to the material science research. Therefore, the following observation are done on the text:
Comment 1: Figure 3: It is suitable to show, in the same graph, the time-temperature curve of the thermocouple used to monitor the furnace interior temperature.
Response: Figure 3 was not produced by the authors. It was taken from a reference.
Comment 2: Review lines 377 and 378, mentioning if the conclusion (8) refers to ambient temperature or high temperature condition. If it is about high temperature, such conclusion is different from the behaviour commented in lines 321 to 329.
Response: Conclusion 8 mentions ductility while comments in lines 321-329 are on strength.
Comment 3: Review the sentence in lines 322 – 323.
Response: Checked and corrected.
Comment 4: Write chapter 7 and References in the same format of the template.
Response: Checked and corrected.
Reviewer 2 Report
1)This reviewer believes that the language of the manuscript needs polishing.
2)The figures are unreadable, please fix this issue
3) In the introduction part, the authors shall bring forward the merit of their study. They should highlight why this study is needed and the shortcomings of the previous studies should be compared.
4) The conclusion part should be rewritten, in its current form, it just summarizes what has been explained before. Discussions related to the conducted parametric study should be added.
5)Please specify the sensor sensitivity, the manufacturer(s) of sensors, and the sensor models
Author Response
Comment 1: This reviewer believes that the language of the manuscript needs polishing.
Response: Checked and corrected.
Comment 2: The figures are unreadable, please fix this issue
Response: The authors rechecked the figures and improved the quality.
Comment 3: In the introduction part, the authors shall bring forward the merit of their study. They should highlight why this study is needed and the shortcomings of the previous studies should be compared.
Response: Checked and corrected.
Comment 4: The conclusion part should be rewritten, in its current form, it just summarizes what has been explained before. Discussions related to the conducted parametric study should be added.
Response: Checked and corrected.
Comment 5: Please specify the sensor sensitivity, the manufacturer(s) of sensors, and the sensor models
Response: Specified.
Round 2
Reviewer 2 Report
I believe the manuscript has been sufficiently improved to warrant publication in Fire